# CRISP: Clustering Multi-Vector Representations for Denoising and Pruning

## Abstract

Multi-vector models, such as ColBERT, are a significant advancement in neural information retrieval (IR), delivering state-of-the-art performance by representing queries and documents by multiple contextualized token-level embeddings. However, this increased representation size introduces considerable storage and computational overheads which have hindered widespread adoption in practice. A common approach to mitigate this overhead is to cluster the model's frozen vectors, but this strategy's effectiveness is fundamentally limited by the intrinsic clusterability of these embeddings. In this work, we introduce CRISP (Clustered Representations with Intrinsic Structure Pruning), a novel multi-vector training method which learns inherently clusterable representations directly within the end-to-end training process. By integrating clustering into the training phase rather than imposing it post-hoc, CRISP significantly outperforms post-hoc clustering at all representation sizes, as well as other token pruning methods. On the BEIR retrieval benchmarks, CRISP achieves a significant rate of **3x** reduction in the number of vectors *while outperforming* the original unpruned model. This indicates that learned clustering effectively denoises the model by filtering irrelevant information, thereby generating more robust multi-vector representations. With more aggressive clustering, CRISP achieves an **11x** reduction in the number of vectors with only a 3.6% quality loss.

## 1 Introduction

Neural embedding models are by now a foundational tool for representing data that underlie the SOTA methods for information retrieval (IR) Zhang et al. (2016), clustering, classification Muennighoff et al. (2022), among many other tasks. Recently, *multi-vector* (MV) representations, introduced by the *late-interaction* framework in ColBERT Khattab & Zaharia (2020), have been shown to deliver significantly improved performance on popular IR benchmarks. ColBERT and its variants Gao et al. (2021); Hofstätter et al. (2022); Lee et al. (2024); Lin et al. (2024); Qian et al. (2022); Santhanam et al. (2022b); Wang et al. (2021); Yao et al. (2021) produce *multiple* embeddings per query or document by generating one embedding per token. The similarity between a query and a document is calculated via so-called *Chamfer Similarity*, also known as the MaxSim operation, between the corresponding sets of vectors. This enables a more fine-grained and expressive representation compared to using a single embedding per query or document, in addition to enabling improved interpretability Formal et al. (2021); Wang et al. (2023) and generalization Lupart et al. (2023); Formal et al. (2022); Zhan et al. (2022); Weller et al. (2023).

Despite these advantages, multi-vector representations are inherently more expensive than single-vector representations. Namely, producing embeddings for every input token increases representation size by multiple orders of magnitude. For instance, on the popular MS MARCO Nguyen et al. (2016) dataset, the canonical ColbertV2 model Santhanam et al. (2022b) produces nearly 80 vectors per document and 32 per query. This increase in scale has many downstream effects. Firstly, it increases the storage requirements for indices which store multi-vector embeddings. Secondly, unlike inner product which is used for single-vector embeddings and has a linear cost in the embedding dimensionality, the runtime of the non-linear Chamfer Similarity scales *quadratically* in the number of embeddings. As a result, the cost and quality of multi-vector retrieval algorithms depend heavily on the number of embeddings Santhanam et al. (2022a); Dhulipala et al. (2024).

The high cost for employing multi-vector embeddings has been a significant barrier to their widespread adoption. Thus, there has been considerable work in recent years to improve the efficiency of multi-vector models Santhanam et al. (2022b); Engels et al. (2024); Hofstätter et al. (2022); Qian et al. (2022); Dhulipala et al. (2024); Clavié et al. (2024); MacAvaney et al. (2025). For instance, the Colbertv2 Santhanam et al. (2022b;a) system and successors employ aggressive centroid-based quantization strategies to reduce index sizes, at the cost of significant complexity and tuning challenges MacAvaney & Tonellotto (2024). An alternative and enticing direction is to *reduce* the number of vectors produced by the model entirely. Several such "pruning" methods have been studied, such as removing vectors which are not as significant to the overall query or passage; this can be done either be learned importance scores or gates Hofstätter

et al. (2022) or by post-hoc pruning of individual vectors based on other mechanisms such as attention weights Liu et al. (2024).

While token-level pruning can be helpful, it has the downside of completely dropping information which fails to exceed a given relevance threshold. Instead, one could hope to learn a holistic representation of the full data with fewer vectors directly. A partial step towards this has been the technique of clustering the multi-vector representations post-hoc, after the model is trained Clavié et al. (2024); Dhulipala et al. (2024). Each cluster is assigned a single pooled vector, which represents the cluster on the aggregate. Unfortunately, this post-hoc approach is limited by the actual clusterability of the frozen embeddings, which were not themselves trained to be clusterable.

In this work, we look beyond post-hoc clustering of multi-vector representations, and study the *learnability* of clustered multi-vector representations. In essence, we question the widespread assumption that token-level embeddings are necessary for the improved expressability of multi-vector models, and instead posit that in fact multi-vector models may have the potential to retain significant expressive capabilities representing data by clustered representations of their token-level embeddings. Specifically, we consider the question:

> *Can multi-vector models be trained to produce inherently clusterable representations with*
> *negligible quality loss?*

### 1.1 Contributions

We introduce CRISP (Clustered Representations with Intrinsic Structure Pruning), a novel training paradigm for multi-vector retrieval models. Unlike prior post-hoc clustering methods that operate on pre-trained, frozen embeddings, CRISP integrates clustering directly into the end-to-end training process. As a result, the model is trained to produce inherently clusterable representations, enabling a significant reduction in representation size with minimal impact on quality, and in some cases even leading to performance improvements. Our main contributions are summarized below.

- **Significant Compression:** We demonstrate that CRISP's significantly improves representation size with minimal drops in quality. Evaluating on the BEIR retrieval benchmark Thakur et al. (2021), our `C8x32` CRISP model surpasses the unpruned multi-vector baseline by 0.4% while compressing document representations by **2.9x** and query representations by **3.9x**. Our more aggressive `C4x8` variant achieves a compression rate of **11x** for documents and **7.9x** for queries with only a 3.6% drop in quality.

- **Denoising Effect:** In addition to compressing representations, we show that CRISP acts as an effective denoising mechanism. By guiding the model to consolidate semantic information by training over clustered representations, CRISP learns to filter out less relevant token-level details, thereby generating more robust representations on datasets prone to noise. For instance, our `C4x8` model outperforms the unpruned model by 5.5% on ArguAna, 6.8% on Scidocs, and 2.7% on NQ, all while reducing the number of document embeddings by 11x. In fact, averaged over all BEIR datasets, our `C8x32` model achieved the top score, *outperforming* even the unpruned model (54.5 vs. 54.3 NDCG@10).

- **Superiority over Post-Hoc Clustering:** CRISP demonstrates a clear advantage over traditional post-hoc clustering techniques. To achieve parity with unpruned models, post-hoc clustering methods achieve only a 2x compression rate limited to *documents only*, whereas our `C8x32` model achieves parity with superior 2.9x document *and* 3.9x query compression rates. At higher compression levels CRISP's benefits are even more pronounced. Our `C4x8` variant, despite its dramatic 11x document and 7.9x query compression rate, experiences only a 3.6% drop in NDCG@10. This is a substantial improvement over post-hoc clustering, which reports a 9.3% degradation limited to a 6x *document-only* compression Clavié et al. (2024). This underscores CRISP's more effective compression-quality trade-off, stemming from its end-to-end training.

## 2 Methodology

This section details our proposed CRISP (Clustered Representations with Intrinsic Structure Pruning) methodology and the baseline pruning techniques against which it is compared. We begin by outlining the challenges with standard multi-vector representations that motivate this work, followed by a description of our base model architecture and the experimental approach to pruning.

### 2.1 Background and Limitations of Multi-Vector Representations

Multi-vector (MV) models learn more expressive representations than traditional single vector models by computing one embedding *per-token* of the input text. This encodes queries and documents as *sets* of vectors $Q, D \subset \mathbb{R}^d$ respectively. Multi-vector then scores the query-document similarity via the Chamfer

Similarity Dhulipala et al. (2024) (also known as MaxSim Khattab & Zaharia (2020)):

$$\text{CHAMFER}(Q, D) = \sum_{q \in Q} \max_{x \in D} \langle q, x \rangle \tag{1}$$

where $\langle \cdot, \cdot \rangle$ is the standard inner product. Beginning with ColBERT Khattab & Zaharia (2020), these multi-vector models have been shown to achieve significant performance improvements over single-vector models. However, due to their increased representation size, there are several key challenges associated with multi-vector models:

- **Computational Expense:** The increased number of vectors per item, combined with the Chamfer scoring that scales quadratically with the number of vectors ($O(MNd)$ for $M$ query and $N$ document vectors), makes MV models computationally expensive and significantly increases their memory footprint.

- **Semantic Redundancy and Skewing Similarity:** Repetitive tokens with similar contextual meaning in the queries can disproportionately affect the Chamfer similarity score, since embeddings of these tokens will appear multiple times in the similarity computation (1). This phenomenon is likely undesirable, since query-document similarity should ideally be based on distinct concepts in the text.

- **Information Noise:** Allocating uniform representational capacity (bits of information) to all tokens, irrespective of their semantic richness, can introduce noise and degrade performance compared to single-vector models in some cases, especially for datasets with long queries (e.g. ArguAna). By clustering of representations, one can hope to mitigate these issues by representing entire regions of the latent space by a single embedding, reducing the effect of outliers and sparsely related embeddings.

Our work, CRISP, makes strides to alleviate all of the above issues by *end-to-end* learning clustered multi-vector representations, thereby **(1)** significantly reducing the number of vectors, improving both computational costs and memory footprint, **(2)** diminishing the effect of redundant query tokens on the similarity, and **(3)** denoising representations by guiding the model towards representations without outliers or spread-clusters.

## 2.2 EXPERIMENTAL FRAMEWORK

**Base Model Architecture:**
The multi-vector models in this work utilize a dual encoder architecture with a `Gemma2B` backbone Team et al. (2024). We finetune `Gemma2B` with the Chamfer similarity loss (1), using the standard methodology employed by ColBERT and models based on ColBERT Khattab & Zaharia (2020); Santhanam et al. (2022b). Building query and document encoders by leveraging pre-trained large language models (LLMs) as encoders has proven to be a strategy that produces high-performing embeddings for textual inputs Ni et al. (2022); Lee et al. (2024). As in Khattab & Zaharia (2020) and Lee et al. (2024), no aggregation or pooling was utilized, and thus our model generates token-level representations. This way, for each token in the textual input the model produces one dense vector representing it. We chose not to project down our representations and kept each vector at the original dimension of 2048. We use contrastive learning based on Chamfer similarity using the large collection of training datasets from Li et al. (2025) that includes publicly available data for retrieval, re-ranking, classification, clustering and sentence-to-sentence similarity (STS). Our training setup includes in-batch random negatives, plus we used the hard negatives included in the retrieval training datasets.

**Training over Pruned Representation**
We will compare CRISP, our clustering-based multi-vector training method, against several fixed-token pruning methods. For all pruned models we consider, the corresponding pruning strategy is applied *during* training of the embeddings; i.e., if we prune to consider only the last 4 tokens, then only these tokens are used in when computing the Chamfer loss during training. This ensures that the model training is aligned with its evaluation. Note that this would not be the case in, for instance, post-hoc clustering of an unpruned model.

The pruning strategies explored in this paper fall into two main types: fixed selection methods, which apply predefined heuristics, and clustering-based approaches, which group vectors by semantic similarity. We now detail the specific strategies within each category.

## 2.3 FIXED-TOKEN PRUNING

We first describe several approaches that select a fixed subset of token vectors using predefined, content-independent heuristics. In all cases, the model is trained with the loss function computed *only* over these selected token vectors, tasking it with learning expressive representations under these constraints.

### 2.3.1 TAIL PRUNING

This method selects only the final $k_q$ token vectors from the query's sequence representation and $k_d$ from the document's, where $k_q$ is usually smaller than $k_d$. The main considerations for this method are twofold: first, for a pre-trained autoregressive language model, the later tokens might capture more summary information of the whole sequence; thus, keeping the later ones might better balance the performance-efficiency trade-off. Second, the query is usually much shorter than the document, and we can choose a smaller $k_q$ for the query but a larger $k_d$ for the document. Since the model is trained over the selected tokens, the goal is for the model to "learn" to move the most relevant information in the text into the embeddings for the last $k_q$ or $k_d$ tokens. We tested two configurations:

- **Tail Pruning (4x8)**: Selects the last 4 query vectors and last 8 document vectors.

- **Tail Pruning (8x32)**: Selects the last 8 query vectors and last 32 document vectors.

If there are fewer tokens in the input than the above fixed size, e.g. less than $k_q$ query tokens or less than $k_d$ document tokenss, then we simply select all tokens to train over.

Embedding Sequence: 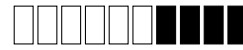

(Example: Keeping last 4 vectors)

Figure 1: Illustration of Tail Pruning, where the last $k$ vectors (here $k = 4$) are retained (shown in black), and others are shown in white.

### 2.3.2 K-SPACING

This method uniformly subsamples the token vectors by selecting every $K$-th vector in the sequence. A key assumption for this method is that adjacent tokens might exhibit similar feature patterns. Consequently, we might perform pruning by sampling at a fixed interval, ensuring the remaining tokens are still evenly distributed throughout the sequence. We tested:

- **K-Spacing (k=4)**: Selects every 4th vector (25% density).

- **K-Spacing (k=2)**: Selects every 2nd vector (50% density).

Embedding Sequence: 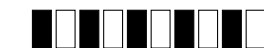

(Example: K-Spacing k=2)

Figure 2: Illustration of K-Spacing, where every $k$-th vector (here $k = 2$) is retained (shown in black), and others are shown in white.

### 2.4 CLUSTERING-BASED PRUNING (THE CRISP APPROACH)

We now describe the clustering-based methods which constitute the CRISP concept. These methods use K-means clustering to group semantically similar token vectors, and then aggregate the clusters into a single vector by using the centroids (i.e., mean-pooling each cluster). By aggregating tokens with similar contextual meanings into a single cluster and representing them with their centroid, CRISP ensures that the contribution comes from distinct semantic units rather than multiple, near-identical token representations. This dynamic grouping also allows for a more flexible allocation of representational capacity, where densely packed semantic regions of the text are captured by individual cluster centroids, while less informative regions might contribute to larger, more general clusters, effectively focusing the model's attention on the most salient aspects of the text.

The key hyperparameter is the choice of $k$, which is the number of clusters and thus the number of vectors used to represent each query or document. We consider two methods of for selecting the hyperparameter $k$.

FIXED-SIZE CLUSTERING

K-means is applied to obtain a pre-defined number of clusters ($k_q$ for query, $k_d$ for document). We tested:

- **Clustering (4x8)**: Uses $k_q = 4$ query centroids and $k_d = 8$ document centroids.

- **Clustering (8x32)**: Uses $k_q = 8$ query centroids and $k_d = 32$ document centroids.

RELATIVE-SIZE CLUSTERING

K-means is applied, but the number of clusters $k$ is set relative to the original sequence length $L$. All resulting centroids are used in scoring. We tested:

- **Clustering (25%)**: $k = \lfloor 0.25 \times L \rfloor$.

- **Clustering (50%)**: $k = \lfloor 0.50 \times L \rfloor$.

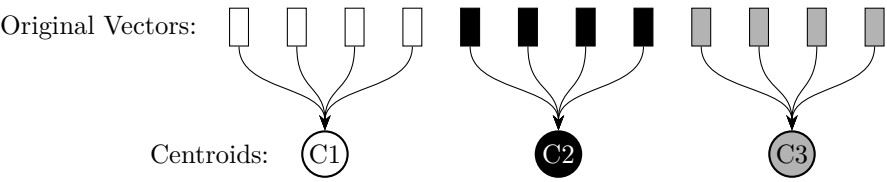

Figure 3: Illustration of Clustering Pruning: Original vectors (top row, styled by assigned cluster) are mapped to their respective cluster centroids (bottom row). Here, 3 clusters (C1-C3) are used, distinguished by white, black, and gray fills. The selected embeddings for each cluster are calculated dynamically so they need not be adjacent.

These methods were evaluated against the baseline single-vector and full multi-vector models, as detailed in the Experiments section.

## 3 EXPERIMENTS

### 3.1 EXPERIMENTAL SETUP

We evaluate CRISP and compare its cluster-based training mechanism with the pruning methods described in Section 2.3. All multi-vector and single-vector models, including the pruned variants, were built upon the `Gemma2B` pre-trained LLM. The models were fine-tuned for 20,000 steps using a batch of size 128 on Cloud TPU v3 [1] using the BGE[2] dataset Li et al. (2025). Crucially, the pruned multi-vector variants (`4x8`, `8x32`, `K4`, `K2`, `C4x8`, `C8x32`, `C25`, `C50`) were fine-tuned starting from the multi-vector ('MV') baseline model.

We evaluate the performance of our models over all the BEIR Thakur et al. (2021) benchmark tasks, a standard suite for evaluating zero-shot retrieval performance. As it is standard for BEIR evaluations, we use NDCG@10 as our metric. We compare our pruned multi-vector models against the standard single-vector ('SV') and multi-vector ('MV') `Gemma2B` models. In addition, when available, we report scores for two other models: GTR$_{xxl}$ Ni et al. (2022), which is a high-quality single-vector model, and XTR$_{xxl}$ Lee et al. (2024), which is a SOTA multi-vector model. Both models are trained from the same T5 backbone Raffel et al. (2020), which differs from our `Gemma2B` backbone, and thus the performance numbers are not directly comparable. However, their inclusion serves to establish a reasonable performance baseline on BEIR.

In all our multi-vector models, including the 'MV' baseline and all pruned variants, inference was conducted using brute-force Chamfer similarity scoring comparing each query against every document in the corresponding dataset corpus. This ensures precise evaluation without confounding factors from approximate retrieval methods.

Given that the base `Gemma2B` model is instruction-tuned, adapting it for the varied information retrieval tasks within the BEIR benchmark required the consistent prepending of task-specific instruction prefixes to the input queries during both the training and inference stages. Initial experiments revealed a substantial decline in retrieval performance when these instructions were absent, suggesting that the model relies on this conditioning to align with the specific retrieval objectives of each task. The precise instruction prefixes utilized for each BEIR task are detailed in Appendix B.

### 3.2 RESULTS AND ANALYSIS

The performance results for all models evaluated on the BEIR tasks are presented in Table 3. The table includes scores for our baseline single-vector ('SV') and multi-vector ('MV') models, the fixed selection pruning methods (Tail Pruning '4x8', '8x32'; K-Spacing 'K4', 'K2'), the clustering-based methods implementing the CRISP approach ('C4x8', 'C8x32', 'C25', 'C50'), and the non-Gemma based models: XTR Lee et al. (2024) and GTR Ni et al. (2022). Tables 1 and 2 show the size of each pruned multi-vector representation relative to the unpruned model for both query and document representations.

---

[1]https://cloud.google.com/tpu/docs/v3

[2]https://huggingface.co/datasets/cfli/bge-full-data/tree/main/data

Table 1: Candidate Representation Size (relative to full MV)

| BEIR Task | Avg Candidate Tokens | Relative Size 4x8, C4x8 | Relative Size 8x32, C8x32 | Relative Size K2, C50 | Relative Size K4, C25 |
|---|---|---|---|---|---|
| arguana | 206.68 | 0.039 | 0.155 | < .50 | < .25 |
| climate_fever | 142.59 | 0.056 | 0.224 | < .50 | < .25 |
| dbpedia_entity | 96.01 | 0.083 | 0.333 | < .50 | < .25 |
| fever | 142.59 | 0.056 | 0.224 | < .50 | < .25 |
| fiqa | 165.34 | 0.048 | 0.194 | < .50 | < .25 |
| hotpotqa | 77.32 | 0.103 | 0.414 | < .50 | < .25 |
| msmarco | 80.74 | 0.099 | 0.396 | < .50 | < .25 |
| nfcorpus | 331.29 | 0.024 | 0.097 | < .50 | < .25 |
| nq | 97.28 | 0.082 | 0.329 | < .50 | < .25 |
| quora | 14.75 | 0.542 | 2.169 | < .50 | < .25 |
| scidocs | 211.06 | 0.038 | 0.152 | < .50 | < .25 |
| scifact | 302.06 | 0.026 | 0.106 | < .50 | < .25 |
| trec_covid | 226.36 | 0.035 | 0.141 | < .50 | < .25 |
| webis_touche2020 | 221.07 | 0.036 | 0.145 | < .50 | < .25 |
| **Average** | **165.37** | **0.091** | **0.349** | **0.50** | **0.25** |

Table 2: Query Representation Size (relative to full MV)

| BEIR Task | Avg Query Tokens | Relative Size for 4x8, C4x8 | Relative Size 8x32, C8x32 | Relative Size K2, C50 | Relative Size K4, C25 |
|---|---|---|---|---|---|
| arguana | 255.48 | 0.016 | 0.031 | < .50 | < .25 |
| climate_fever | 44.31 | 0.090 | 0.181 | < .50 | < .25 |
| dbpedia_entity | 23.75 | 0.168 | 0.337 | < .50 | < .25 |
| fever | 28.44 | 0.141 | 0.281 | < .50 | < .25 |
| fiqa | 31.40 | 0.127 | 0.255 | < .50 | < .25 |
| hotpotqa | 39.88 | 0.100 | 0.201 | < .50 | < .25 |
| msmarco | 25.21 | 0.159 | 0.317 | < .50 | < .25 |
| nfcorpus | 21.41 | 0.187 | 0.374 | < .50 | < .25 |
| nq | 25.97 | 0.154 | 0.308 | < .50 | < .25 |
| quora | 31.32 | 0.128 | 0.255 | < .50 | < .25 |
| scidocs | 33.31 | 0.120 | 0.240 | < .50 | < .25 |
| scifact | 35.98 | 0.111 | 0.222 | < .50 | < .25 |
| trec_covid | 29.82 | 0.134 | 0.268 | < .50 | < .25 |
| webis_touche2020 | 25.86 | 0.155 | 0.309 | < .50 | < .25 |
| **Average** | **46.58** | **0.128** | **0.256** | **0.50** | **0.25** |

Tables 1 and 2 show the average relative representation size, in terms of number of output embeddings, for each pruned multi-vector model compared to the unpruned model. Thus, the unpruned model has a relative size of 1, whereas C50, which outputs half as many of the embeddings, will have a relative size of at most 50%. An average relative size of .091 thus corresponds to a compression rate of $1/.091 \approx 11$.

Table 3: Pruning in BEIR (NDCG@10)

| BEIR Task | External Baselines | | Gemma 2B-based Models | | | | | | | | | |
| | | | Unpruned | | Fixed Selection | | | | CRISP | | | |
| | XTR | GTR | MV | SV | 4x8 | 8x32 | K4 | K2 | C4x8 | C8x32 | C25 | C50 |
|---|---|---|---|---|---|---|---|---|---|---|---|---|
| arguana | 44.2 | 54.0 | 70.1 | 73.8 | 53.6 | 56.4 | 67.4 | 69.3 | **74.0** | 71.6 | 66.4 | 69.2 |
| climate_fever | 24.5 | 26.7 | 39.6 | 35.7 | 8.1 | 10.3 | 35.7 | **40.0** | 36.0 | 36.0 | 28.4 | 31.3 |
| dbpedia_entity | **44.3** | 40.8 | 42.1 | 35.4 | 17.1 | 29.3 | 32.4 | 40.4 | 37.5 | 38.5 | 39.2 | 37.8 |
| fever | 77.0 | 74.0 | **90.3** | 82.8 | 28.5 | 34.3 | 87.4 | 89.6 | 87.3 | 83.5 | 83.5 | 88.4 |
| fiqa | 43.8 | 46.7 | 49.4 | 45.6 | 16.1 | 25.2 | 42.0 | 47.2 | 47.4 | **50.1** | 46.6 | 49.2 |
| hotpotqa | 66.2 | 59.9 | **72.6** | 59.2 | 16.1 | 25.9 | 62.7 | 70.6 | 68.5 | 70.1 | 69.0 | 70.6 |
| msmarco | **46.6** | 44.2 | 25.9 | 39.9 | 13.7 | 17.3 | 23.7 | 29.4 | 41.7 | 42.9 | 42.1 | 43.3 |
| nfcorpus | 35.3 | 34.2 | **36.4** | 31.8 | 16.4 | 27.8 | 33.5 | 36.0 | 32.3 | 36.0 | 34.3 | 36.2 |
| nq | 60.9 | 56.8 | 62.0 | 56.8 | 27.5 | 40.0 | 52.7 | 62.5 | 63.7 | **65.2** | 62.6 | **65.2** |
| quora | 88.1 | 89.2 | 87.7 | 87.4 | 63.7 | 81.4 | 80.4 | 86.1 | 89.1 | **89.3** | 88.6 | 89.2 |
| scidocs | 17.1 | 16.1 | 21.9 | 18.8 | 8.8 | 13.7 | 18.1 | 21.7 | 23.4 | 23.2 | 20.9 | **23.8** |
| scifact | **74.3** | 66.2 | 73.8 | 49.5 | 21.6 | 36.3 | 65.6 | 72.3 | 55.5 | 65.8 | 45.7 | 58.9 |
| trec_covid | **78.9** | 50.1 | 63.0 | 53.5 | 22.8 | 29.0 | 54.4 | 60.5 | 52.0 | 63.2 | 50.6 | 48.0 |
| webis_touche2020 | **30.9** | 23.3 | 25.0 | 28.8 | 8.5 | 18.6 | 22.0 | 24.6 | 23.7 | 27.3 | 25.6 | 24.4 |
| Total | 52.7 | 49.1 | 54.3 | 49.9 | 23.0 | 31.8 | 48.4 | 53.6 | 52.3 | **54.5** | 50.3 | 52.5 |
| Avg Rel Doc Size | — | — | 1 | — | .091 | .349 | .25 | .50 | .091 | .349 | .25 | .50 |
| Avg Rel Query Size | — | — | 1 | — | .128 | .256 | .25 | .50 | .128 | .256 | .25 | .50 |
| cqadupstack | — | — | 38.7 | 34.7 | 10.8 | 8.9 | 31.2 | 37.1 | 38.6 | **42.1** | 36.3 | 41.7 |
| cq_android | — | — | 45.6 | 43.4 | 20.6 | 16.7 | 39.3 | 44.7 | 46.5 | **50.6** | 47.6 | 50.4 |
| cq_english | — | — | 47.2 | 42.3 | 12.0 | 10.1 | 40.3 | 45.3 | 46.1 | **49.5** | 44.0 | 48.7 |
| cq_gaming | — | — | 50.2 | 49.9 | 10.3 | 3.9 | 40.3 | 48.7 | 54.6 | 57.9 | 51.2 | **59.3** |
| cq_gis | — | — | 34.4 | 28.2 | 6.4 | 4.5 | 25.8 | 32.6 | 34.7 | **39.8** | 31.9 | 37.8 |
| cq_mathematica | — | — | 28.7 | 24.4 | 9.9 | 8.5 | 22.7 | 28.0 | 26.2 | **31.8** | 23.5 | 29.3 |
| cq_physics | — | — | 44.1 | 42.4 | 15.5 | 16.0 | 36.9 | 43.1 | 44.2 | 46.4 | 42.6 | **47.7** |
| cq_programmers | — | — | 41.0 | 39.6 | 9.8 | 7.4 | 35.8 | 40.1 | 41.4 | 42.9 | 38.0 | **43.4** |
| cq_stats | — | — | 35.8 | 30.1 | 7.6 | 5.2 | 26.5 | 33.4 | 32.9 | 36.3 | 30.9 | **36.6** |
| cq_tex | — | — | 27.6 | 22.0 | 6.2 | 5.2 | 20.0 | 25.7 | 26.0 | **30.3** | 22.0 | 27.9 |
| cq_unix | — | — | 42.8 | 32.5 | 12.2 | 11.8 | 35.2 | 41.3 | 39.0 | **43.5** | 37.5 | 42.2 |
| cq_webmasters | — | — | 39.7 | 36.4 | 13.8 | 13.3 | 32.9 | 37.8 | 38.9 | 41.6 | 38.1 | **43.2** |
| cq_wordpress | — | — | 26.8 | 24.7 | 4.7 | 4.0 | 18.5 | 24.8 | 33.1 | **35.0** | 28.7 | 34.0 |

Consistent with prior findings, the baseline 'MV' model (Avg 54.3) significantly outperforms the baseline 'SV' model (Avg 49.9), demonstrating the general advantage of multi-vector representations. Fixed selection pruning methods, however, lead to a substantial degradation in performance. Both Tail Pruning variants ('4x8': 23.0, '8x32': 31.8) and K-Spacing ('K4': 48.4, 'K2': 53.6) score considerably lower than the 'MV' baseline on average. Even 'K2', which retains 50% of the vectors, does not fully recover the baseline performance.

In contrast, the CRISP models prove much more effective. CRISP is significantly better than fixed selection in almost all BEIR tasks, particularly when compared to methods like '4x8' or '8x32' which consider only a small, fixed positional subset of vectors. The best performing model overall is the CRISP 'C8x32' model (Avg 54.5), which slightly surpasses the performance of the full 'MV' baseline. Other CRISP variants like 'C4x8' (52.3) 'C25' (50.2) and 'C50' (52.5) also remain competitive and substantially outperform the fixed selection models. This suggests that K-means clustering effectively identifies and retains the most salient semantic information within the multiple token embeddings. Further, we observed that CRISP can indeed improve upon the vanilla multi-vector baseline in several cases (e.g., ArguAna 'C4x8' vs 'MV'; FIQA 'C8x32' vs 'MV'; NQ 'C8x32'/'C50' vs 'MV'; Quora 'C8x32' vs 'MV'), potentially due to a denoising effect where less informative token variations within clusters are abstracted away by the centroid representation (discussed in Appendix A). The overall strong performance of 'C8x32' indicates that the CRISP approach provides a viable path towards reducing the complexity of multi-vector models while maintaining, and sometimes enhancing, retrieval effectiveness.

It is important to note that the effectiveness of all the models presented were found to be highly sensitive to hyperparameter tuning. Specifically, the learning rate and L2 weight normalization played critical roles in achieving the reported results. Furthermore, as detailed in Appendix B, the practice of prepending task-specific instruction prefixes to the input queries during inference proved indispensable. Omitting or poorly configuring these elements significantly degraded retrieval performance, underscoring that the presented qualitative examples and quantitative successes are contingent upon careful optimization of these crucial training and inference parameters.

**Comparison to Post-Hoc Clustering.** A primary alternative for comparison is *post-hoc* clustering, where clustering is applied to a multi-vector model after its embeddings are frozen. This approach offers a simpler training pipeline as it avoids clustering during the training phase, making it a natural baseline. For example, post-hoc clustering was employed to enhance retrieval latency in Dhulipala et al. (2024), and is the main focus of Clavié et al. (2024), which explores clustering methods such as k-means and Hierarchical Clustering for pruning multi-vector representations. They found Hierarchical clustering to be the best performing method when evaluated over a subset of the BEIR retrieval tasks. Specifically, they showed that, compared to the unpruned multi-vector model, this method yields: a 2x compression with a 0.6% NDCG@10 improvement, a 4x compression with a 3% NDCG@10 decrease, and a 6x compression with a 9.3% NDCG@10 decrease. Notably, in Clavié et al. (2024) the authors applying pruning to the document representations only, leaving the query embeddings unchanged.

CRISP's cluster-based training, however, presents considerably more favorable compression-quality trade-offs. The C8x32 CRISP model, for instance, achieves **2.9x** document token compression, in addition to a **3.9x** query token compression, while improving NDCG@10 by 0.4%. Furthermore, the more aggressive C4x8 CRISP model delivers **11x** document token and **7.9x** query token compression with only a 3.6% decrease in NDCG@10. Therefore, CRISP not only yields better document compression rates with comparable or superior quality retention than post-hoc clustering, but crucially also enables substantial query-token compression simultaneously — a benefit not achieved in Clavié et al. (2024). We emphasize that compressing query tokens has significant impact on downstream task, such as reducing retrieval latency, as demonstrated in Dhulipala et al. (2024) where pre-clustering query tokens markedly sped up retrieval, and in Santhanam et al. (2022a) where each query vector necessitates a separate vector-index query.

## 4 OTHER RELATED WORK

The seminal ColBERT model Khattab & Zaharia (2020) introduced multi-vector models as a way to improve passage and document search via token-level interactions and representations of textual inputs. Since then, significant effort has been devoted to developing improved and optimized multi-vector models and retrieval methods Santhanam et al. (2022b;a); Gao et al. (2021); Hofstätter et al. (2022); Lee et al. (2024); Lin et al. (2024); Qian et al. (2022); Santhanam et al. (2022b); Wang et al. (2021); Yao et al. (2021); Dhulipala et al. (2024). Nevertheless, multi-vector models still require non-trivial computational overheads when compared to single-vector models. To address these efficiency issues, MacAvaney et al. (2025) use a fixed number of vectors irrespective of the length of the queries or the documents and demonstrated that they are able to retain the performance of ColBERT-v2 Santhanam et al. (2022b) after reduction. In contrast, Lee et al. (2024) trained the multi-vector model to purposely use during retrieval the vectors that represent the most salient parts of the queries and the documents. ALIGNER Qian et al. (2022)

introduced linear programming for sparse query-document alignments and per token-salience. ColBERTer Hofstätter et al. (2022) followed up with a vector reduction approach based on pooling embeddings of tokens within the same word, as well as pruning by removing stop words. However, this pooling is fixed in advance and not learned based on the neural representations of the input tokens. In contrast to these prior approaches, CRISP is the first method to learn a pooling (i.e. clustering) of the token representations during training time, allowing the model to couple the learning of the representations to the learning of the pooling.

## 5 CONCLUSION

In this paper, we introduced CRISP (Clustered Representations with Intrinsic Structure Pruning), a novel multi-vector model method that learns inherently clusterable representations during end-to-end training, thereby significantly reducing the representation size of multi-vector models. CRISP produces pruned models that actually *outperform* the original unpruned models, while compressing document representations by **3x**, as well as offering an improved **11x** compression at the cost of a small drop in performance. By learning clusters during training, CRISP also significantly outperforms post-hoc clustering methods that operate on frozen embeddings. Thus, CRISP offers a significant step towards bridging the efficiency gap between multi-vector and single-vector models.

**Broader Impacts and Limitations:**

Our work primarily focuses on improving the quality-efficiency trade-off of neural information retrieval (IR) systems. Improved quality of IR systems has the potential benefit of improving user experience by improving the quality of IR queries. While search products themselves may have some negative societal impacts, it is unlikely that our work will have any direct path to negative applications or an affect on these impacts. As for limitations, a primary limitation CRISP is that it fixes the maximum number of clusters $k$ to be used during training in advance. This prevents the model from learning the optimal number of clusters to use for a given query or document, which is a benefit of unpruned multi-vector models. We leave the exploration of methods which adapt the number of clusters, as well as the exploration of alternative clustering mechanisms, to future work.

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

## A   APPENDIX: QUALITATIVE CLUSTERING EXAMPLES

To offer qualitative insights into the behavior of K-means clustering on token embeddings, this appendix presents two examples selected from the ArguAna dataset Wachsmuth et al. (2018), where the task is to retrieve a counterargument for a given argument query. Each example includes the query text, the corresponding gold counterargument document, with some highlighted tokens that belong to the same cluster. In each case, CRISP was able to retrieve the top matching document. The matching highlighted cluster in the document reflects the document cluster that achieved the highest multi-vector similarity (Chamfer) with that particular query cluster. These visual examples highlight a recurring pattern: K-means clustering frequently isolates semantically lighter tokens, such as stopwords, punctuation, and generic terms, into distinct clusters.

**Task Description**   Given a claim, find documents that refute the claim.

### A.1   EXAMPLE 1

**Query Text**

Given a claim , find documents that refute the claim . | query : A UN standing army is simply impossible to form . A standing army for the United Nations has an existing legal framework ; it has never been attempted in practice because it would be impossible to create . Article 4 3 of the original UN Charter specifies that all member states are expected , upon the signing of a future UN agreement , to provide 'forces , assistance and facilities ' for the maintenance of international peace and security 1 . That it is has never been attempted is the direct result of its sheer impracticality ; who would contribute the troops ? How would they be trained , and ensure that troops trained in one state would not be asked to thereafter fire on their own colleagues ? Furthermore , where would the U . N. standing army be located , for the United Nations has no land , and the United States would not take kindly to a reprisal attack on the UN Army at the United Nations Headquarters . And who would fund this army ? The United States hasn ' t paid its bills to the United Nations in years due to their opposition to some of its actions / What is there in place to prevent that continuing ? Lastly , and most importantly , whose will would they be implementing , for the United Nations is not a single voice but the aggregated noise of its member states ? The Security Council , which currently dictates the form that U . N. peacekeeping operations take , are not a group to whom impartiality can be attributed . A U . N . standing army at the be hest of the Security Council would be used sparingly at best and only in regions and conflicts for whom all the P5 had a vested interest in the maintenance of peace . Any impartiality that the U . N . standing army had in theory would be lost in practice . 1 . U . N. Charter , ( 1 945)

**Document Text (Refutation)**

global politics defence war peace house would create UN standing army A U . N . standing army is not impossible to form . The United Nations has already conclusively proved , in numerous peacekeeping among other missions , its ability to play a constructive , effective military role in interventions ; a standing army would merely replace the top level of command . Instead of taking orders from the top brass in a national military , the orders would come from United Nations commanders . For soldiers trained to listen and respond to commands , this would constitute merely a subtle shift that would not alter their operational effectiveness . Furthermore , funding would be provided through similar streams to how peacekeeping forces are funded contemporaneously ;; however , once the U . N . standing army has proved itself capable , funding will surely come from those states who recognize that pooling resources to form a U . N . army is more prudent than scratching together a under-resourced , native army .

## A.2 EXAMPLE 2

**Query Text**

Given a claim , find documents that refute the claim . | query : The law is hypocritical In most countries where drugs are illegal , tobacco and alcohol , which arguably have equally devastating consequences in society , are legal . In a UK study , alcohol was shown to have the worst effects of any drug , yet the current law recognises that people should be able to choose whether they drink or not . [ 1 ] The same should be true of drugs . [ 1 ] Professor David Nutt , ' Drug Harms in the UK : a multicriteria decision analysis ', The Lancet , Vol 376 , Issue 9752 , pp . 1558-1565 , 6th November 2010 ,

**Document Text (Refutation)**

th addiction health general law crime policing house supports legal isation drugs Perhaps alcohol and tobacco should also be illegal . However , one of the reasons why alcohol ranks so badly in such studies is because of its legality ; if other drugs were legal , we would see their usage go up and therefore the negative social effects they produce rise as well .

## A.3 DISCUSSION OF CLUSTERING EXAMPLES

The examples presented (Example 1, Example 2) provide a qualitative view into the behavior of K-means clustering on token embeddings. A recurring observation is that certain clusters tend to aggregate tokens with lower semantic weight, such as punctuation, common stopwords (e.g., 'a', 'the', 'of', 'and', 'that', 'would'), or formatting elements (e.g., '|', ',', '.', ';', '?'). For instance, in Example 1, the highlighted cluster contains mostly punctuation, single letters, and generic query/instruction tokens like 'Given', 'find', 'query'. In the corresponding Document Text, the matching cluster also consists mostly of stop words and punctuation.

Note that the prompt prefix used in this particular dataset is prepended to the queries, and the document title, consisting of buzz words provided with the dataset, is prepended to the documents. These instruction words are usually grouped together in a single cluster or together with other semantically light tokens (consider the highlighted cluster in Example 2 and the matching document cluster).

This behavior aligns with the hypothesis behind CRISP: that clustering can serve as a denoising mechanism. During the Chamfer similarity (i.e. MaxSim) calculation inherent in multi-vector retrieval, query clusters dominated by these low-content tokens are less likely to find strong matches within the document's token embeddings, or more precisely these clusters will show no strong preference for one document over another, effectively reducing their contribution to the final similarity score. In the document representation, these clusters often group similarly generic tokens. As Chamfer focuses on the *maximum* similarity for each query token (or centroid), document clusters containing only stopwords are less likely to be the maximal match for query clusters containing substantive terms. Consequently, the influence of these ubiquitous but often irrelevant tokens is mitigated in both the query and document representations, potentially leading to a clearer signal for relevance matching based on more meaningful terms, which helps explain why CRISP (via these trained clustering configurations) can sometimes outperform the full multi-vector baseline. If we removed these tokens altogether from the similarity computation we would likely find even a further increase in performance.

## B    APPENDIX: TASK INSTRUCTION PREFIXES FOR BEIR DATASETS

The base model used in our experiments, `Gemma2B`, is primarily instruction-tuned for conversational tasks. To adapt this model effectively for the diverse retrieval tasks within the BEIR benchmark, we found it necessary to prepend task-specific instruction prefixes to the input queries during both training and inference. Preliminary experiments indicated that omitting these instructions significantly degraded retrieval performance, likely due to the model not being optimally conditioned for the specific retrieval goal of each dataset.

The instruction prefixes in Table 4 were used for each corresponding BEIR dataset.

Table 4: Task Instruction Prefixes for BEIR Datasets

| Dataset | Instruction Prefix |
| --- | --- |
| `arguana` | *Given a claim, find documents that refute the claim.* |
| `climate-fever` | *Given a claim about climate change, retrieve documents that support or refute the claim.* |
| `dbpedia-entity` | *Given a query, retrieve relevant entity descriptions from DBPedia.* |
| `fever` | *Given a claim, retrieve documents that support or refute the claim.* |
| `fiqa` | *Given a financial question, retrieve user replies that best answer the question.* |
| `hotpotqa` | *Given a multi-hop question, retrieve documents that can help answer the question.* |
| `msmarco` | *Given a web search query, retrieve relevant passages that answer the query.* |
| `nfcorpus` | *Given a question, retrieve relevant documents that best answer the question.* |
| `nq` | *Given a question, retrieve Wikipedia passages that answer the question.* |
| `quora` | *Given a question, retrieve questions that are semantically equivalent to the given question.* |
| `scidocs` | *Given a scientific paper title, retrieve paper abstracts that are cited by the given paper.* |
| `scifact` | *Given a scientific claim, retrieve documents that support or refute the claim.* |
| `trec-covid` | *Given a query, retrieve documents that answer the query.* |
| `webis-touche2020` | *Given a question, retrieve detailed and persuasive arguments that answer the question.* |
| `cqadupstack` | *Given a question, retrieve detailed question descriptions from Stackexchange that are duplicates to the given question.* |

Applying these specific instructions helps align the conversational base model with the target retrieval task for each dataset.

## C    APPENDIX: DATASET LICENSES

In the main body of the paper we cite references for both our training Li et al. (2025) and evaluation Thakur et al. (2021) datasets. We include the URL where the training datasets are available. Regarding the evaluation datasets, we refer to the licensing information disclosed in page 20 of Thakur et al. (2021). Regarding the training dataset, our investigation about the licensing of the BGE training data shows the following heterogeneous licensing overview:

- SQuAD: CC BY-SA 4.0.[3]

---

[3]https://rajpurkar.github.io/SQuAD-explorer/

- FEVER: Licensed by a combination of Wikipedia Copyright Policy and CC-BY-SA 3.0.[4]

- NQ: Provided under CC BY-SA 3.0 license.[5]

- NLI - SNLI under CC BY-SA 4.0,[6] MNLI under a combination of the OANC's license, CC BY-SA 3.0, CC BY 3.0[7] with preparation code licensed under MIT license[8]

- FiQA: Unknown.[9]

- Emotion-Classification educational and research use only.[10]

- MTOPIntent-Classification CC BY-SA 4.0.[11]

- StackOverFlowDupQuestions (LinkSO) Unknown.[12]

- ArguAna,[13] SciDocsRR,[14] Banking77-Classification[15]: Provided under CC BY 4.0 license.

- ArxivClustering annotations licensed under CC0[16] with content under one of the following: CC BY-SA 4.0, CC BY-NC-SA 4.0, CC BY-NC-ND 4.0, arXiv.org perpetual, non-exclusive license 1.0, or CC Zero.[17]

- Biorxiv content under one of the following: no reuse/adaptation without permission, CC-BY-NC-ND, CC-BY-ND, CC-BY-NC, CC-BY, or CC0.[18]

- Medrxiv content under one of the following: no reuse/adaptation without permission, CC-BY-NC-ND, CC-BY-ND, CC-BY-NC, CC-BY, or CC0.[19]

- RedditClustering and StackExchangeClustering Unknown.[20]

- TwentyNewsgroupsClustering CC BY 4.0.[21]

- STS12 Unknown/mixed with some data licensed under Microsoft Research licenses.[22]

- TriviaQA Unknown - passages sourced from web documents and Wikipedia.[23]

- AmazonCounterfactualClassification CC BY-SA 4.0.[24]

- TweetSentimentExtractionClassification CC BY 4.0.[25]

- IMDB-Classification Unknown.[26]

- ToxicConversationsClassification cc0-1.0[27]

- ELI5: Unknown - harvested from Reddit comments.[28]

- HotpotQA: Provided under the CC BY-SA 4.0 license.[29]

---

[4]https://fever.ai/download/fever/license.html
[5]https://ai.google.com/research/NaturalQuestions/download
[6]https://nlp.stanford.edu/projects/snli/
[7]https://huggingface.co/datasets/nyu-mll/multi_nli
[8]https://github.com/princeton-nlp/SimCSE
[9]https://sites.google.com/corp/view/fiqa/home
[10]https://github.com/dair-ai/emotion_dataset
[11]https://fb.me/mtop_dataset link downloads dataset.
[12]https://sites.google.com/corp/view/linkso
[13]https://zenodo.org/records/3973258
[14]https://github.com/allenai/scidocs/blob/master/LICENSE
[15]https://github.com/PolyAI-LDN/task-specific-datasets
[16]https://www.kaggle.com/datasets/Cornell-University/arxiv
[17]https://info.arxiv.org/help/license/index.html
[18]https://www.biorxiv.org/about/FAQ
[19]https://www.medrxiv.org/about/FAQ
[20]https://github.com/UKPLab/TWEAC-qa-agent-selection
[21]https://archive.ics.uci.edu/dataset/113/twenty+newsgroups
[22]https://web.archive.org/web/20201029123711/https://www.cs.york.ac.uk/semeval-2012/task6/, http://ixa2.si.ehu.eus/stswiki/
[23]https://nlp.cs.washington.edu/triviaqa/
[24]https://github.com/amazon-science/amazon-multilingual-counterfactual-dataset
[25]https://www.kaggle.com/competitions/tweet-sentiment-extraction/overview
[26]http://ai.stanford.edu/ amaas/data/sentiment/index.html
[27]https://huggingface.co/datasets/google/jigsaw_unintended_bias
[28]https://facebookresearch.github.io/ELI5/
[29]https://hotpotqa.github.io/

- AmazonReviewsClassification: academic research use dataset license.[30]

- Quora Duplicate Questions Detection: Unknown.[31]

- MSMARCO passage and document ranking is distributed for non-commercial resource purposes by Microsoft, but not extending any license or other intellectual property rights.[32]

- STS22: Unknown - harvested news articles.[33]

- STSBenchmark: Unknown - aggregation of datasets from SemEval STS shared tasks from 2012-2017.[34]

---

[30]https://github.com/awslabs/open-data-docs/tree/main/docs/amazon-reviews-ml
[31]https://www.kaggle.com/c/quora-question-pairs
[32]https://microsoft.github.io/msmarco/Datasets.html
[33]https://competitions.codalab.org/competitions/33835
[34]http://ixa2.si.ehu.eus/stswiki/

