# OpenReview forum: "CRISP: Clustering Multi-Vector Representations for Denoising and Pruning"
_ICLR.cc/2026/Conference — ICLR 2026 Conference Withdrawn Submission_

### Official Review · Reviewer_Thz6 · 2025-11-01

**Soundness:** 1
**Presentation:** 2
**Contribution:** 2
**Rating:** 4
**Confidence:** 4

**Summary:**

In information retrieval (IR) tasks, the efficiency of multi-vector retrieval is heavily influenced by the number of vectors used. This paper introduces CRISP (Clustered Representations with Intrinsic Structure Pruning), which aims to improve the clustering efficiency of multi-vector retrieval models. CRISP learns clusterable representations end-to-end during training, unlike post-hoc clustering or token pruning methods that operate on frozen embeddings. Experiments were conducted on the BEIR benchmark using the Gemma2B model.

**Strengths:**

- The paper presents the idea of integrating clustering into the training process, enabling the model to learn representations that are inherently more suitable for clustering.

- Overall, the paper is clearly written and easy to follow.

**Weaknesses:**

- Insufficient baseline comparison: The authors only compare their method with other clustering-based approaches and very simple fixed-token pruning methods. However, recent research has actively explored dynamic token and representation compression techniques that enhance efficiency (e.g., [1]). Such methods may in fact address the three key challenges listed in lines 125–139 more directly, by compressing redundancy while minimising information loss, thereby improving computational efficiency. If the paper compares CRISP against token pruning methods as baselines for efficient IR, it should also compare with dynamic token compression approaches.

- Limited model diversity: Aside from the external baselines, both the proposed and baseline models appear to rely solely on Gemma2B. Since CRISP is designed as an end-to-end training method, demonstrating that it generalises across different model architectures would strengthen its contribution, particularly given that the authors note the method is highly sensitive to hyperparameter tuning (lines 425–426).

- Minor editing issues: When a citation functions as part of a sentence, the paper should use \citep{} (see ICLR 2026 formatting instructions, Section 4.1). The description of Tables 1 and 2 (lines 286–287) is duplicated later in lines 338–339.


[1] Nawrot, Piotr, et al. “Dynamic Memory Compression: Retrofitting LLMs for Accelerated Inference.” International Conference on Machine Learning. PMLR, 2024.

**Questions:**

- As mentioned in the weaknesses, could the authors compare CRISP with dynamic token compression methods?

- Could the authors apply the proposed training approach to models other than Gemma2B to demonstrate broader generalisability?

- In lines 431–432, the comparison to post-hoc clustering is discussed but does not appear in Table 3. Was there a reason why those methods could not be evaluated under the same experimental setting in Table 3?

---

### Official Review · Reviewer_m7WV · 2025-11-04

**Soundness:** 3
**Presentation:** 2
**Contribution:** 3
**Rating:** 6
**Confidence:** 4

**Summary:**

The authors propose a method for training multi-vector retrieval methods to produce naturally "clusterable" vector representations, to enable reducing the number of vector embeddings per document (or query) to be as little as 3-11x fewer than the standard approach, while seeking to preserve (or even slightly improve) quality.

To do so, the authors apply k-means clustering per sequence (query or document) during encoding at training time. Each sequence is thus represented as a set of the average of all vectors per cluster, instead of one vector per token. Contrastive-esque training proceeds as usual otherwise, with the Chamfer/MaxSim similarity.

This method (CRISP) is applied to fine-tuning Gemma2-based multi-vector retrievers. The CRISP model is competitive with the multi-vector baseline while compressing document representations by 3x; more aggressive compression (11x) is possible at only a 3-4% drop in quality. As the authors argue, this is indeed a substantial improvement over pre-existing "post-hoc" clustering methods, which degrade more quickly even with less aggressive compression.

**Strengths:**

The method proposed is very simple (and hence likely to be picked up and generate impact) and quite effective, as shown by the results on BEIR. The authors consider a number of reasonable baselines and train all models in a fair manner. This could enable a fair and scientific comparison, though it has the disadvantage of being somewhat disconnected from the broader literature on training techniques for many of these models.

**Weaknesses:**

The paper offers limited insight about the cost or complexity of running k-means clustering during training. How expensive or complex is this? Were any special tricks necessary for dealing with the gradient propagation? Why wasn't it done before, at least so effectively? Are there any theoretical or conceptual concerns that should be considered in clustering _per document_ versus clustering across the corpus?

BEIR is a fairly old and "easy" / rather statured benchmark at this point. What about the various more modern IR benchmarks, e.g. like BRIGHT?

**Questions:**

See Weaknesses.

---

### Official Review · Reviewer_6Wpt · 2025-11-05

**Soundness:** 2
**Presentation:** 2
**Contribution:** 2
**Rating:** 2
**Confidence:** 4

**Summary:**

The paper proposes CRISP, an approach that integrates clustering into the training process of multi-vector retrieval models to learn inherently clusterable token representations. Instead of applying post-hoc clustering or token pruning, CRISP encourages structured representation learning via a clustering loss (Chamfer distance) and a pruning mechanism that discards redundant tokens. Experiments on the BEIR benchmark aim to show that CRISP achieves competitive retrieval performance with significantly fewer tokens, suggesting that learned clusterable representations can improve both efficiency and robustness.

**Strengths:**

1. The paper tackles an important and practical problem in neural information retrieval: the efficiency and redundancy of token-level multi-vector representations.

2. The experimental section is  covering multiple BEIR tasks and including comparisons with  multi-vector and single-vector models, including the pruned variants.

**Weaknesses:**

1. The main concern with this paper lies in its lack of substantial methodological innovation. While CRISP presents a training paradigm that integrates clustering directly into the end-to-end learning process of multi-vector retrieval models, this idea is not fundamentally new. Prior works, such as “Deep Clustering for Unsupervised Learning of Visual Features” and subsequent extensions, have already explored integrating k-means-style clustering objectives into representation learning frameworks. In the context of information retrieval, several studies have similarly incorporated clustering or token grouping mechanisms during training to achieve efficiency gains.
Therefore, the paper’s main contribution appears incremental, and it remains unclear what is technically novel beyond applying a well-known clustering regularization idea to multi-vector retrieval. The authors are encouraged to clearly articulate the unique algorithmic contributions or theoretical insights that distinguish CRISP from existing clustering-based compression or pruning methods.

2. The paper provides limited understanding of why clustering improves performance or efficiency. Although empirical results suggest that CRISP can act as a denoising mechanism, the analysis of this phenomenon is superficial. There is no visualization or quantitative exploration of the learned cluster structures, token distribution changes, or training dynamics that might explain how clusterable representations enhance retrieval robustness.

3. The experimental evaluation compares CRISP primarily against simple fixed-selection pruning methods, which are relatively outdated and limited in scope. This makes it difficult to position CRISP fairly within the growing body of work on efficient multi-vector and token pruning techniques. Many recent approaches have investigated learnable pruning, dynamic token reduction, and attention-based selection mechanisms that offer stronger baselines for comparison. Evaluating CRISP against these state-of-the-art pruning and compression methods would provide a more convincing demonstration of its advantages and clarify its practical relevance.

4. The paper does not fully adhere to the ICLR formatting and style guidelines, which affects its overall readability and professionalism.

**Questions:**

Please see above weaknesses.

---

### Note · Authors · 2025-12-03

I have read and agree with the venue's withdrawal policy on behalf of myself and my co-authors.